# Macroscopic Properties and Pore Structure Fractal Characteristics of Alkali-Activated Metakaolin–Slag Composite Cementitious Materials

**DOI:** 10.3390/polym14235217

**Published:** 2022-11-30

**Authors:** Jianghuai Zhan, Bo Fu, Zhenyun Cheng

**Affiliations:** 1School of Civil Engineering, North Minzu University, Yinchuan 750021, China; 2National Energy Group, Coal Chemical Industry Technology Research Institute, Ningxia Coal Industry Co., Ltd., Yinchuan 720021, China

**Keywords:** alkali-activated metakaolin–slag composite cementitious material, mechanical properties, drying shrinkage, microscopic analysis, fractal dimension

## Abstract

To investigate the effects of slag and Na_2_O content on the macroscopic properties and pore structure characteristics of alkali-activated metakaolin–slag (AAMS) composite cementitious materials, this study used X-ray diffraction (XRD), Fourier transform infrared spectroscopy (FT-IR), scanning electron microscopy (SEM-EDS), and mercury-pressure (MIP) tests for characterization and analyzed the hydration product compositions, microstructures, and pore structure characteristics of AAMS composite cementitious materials. The relationships between the fractal dimension and the pore structure parameters, compressive strengths, and drying shrinkage rates of AAMS composite cementitious materials were investigated with the thermodynamic fractal model. The results showed that at the age of 28 d, the compressive strength and drying shrinkage of the AAMS composite binder increased by 20.57% and 215.11%, respectively, when the slag content increased from 0 to 50%. When the Na_2_O content increased from 8% to 12%, the compressive strength and drying shrinkage of the AAMS composite increased by 24.37% and 129.40%, respectively. The compressive strengths of AAMS composite cementitious materials increased with increasing slag content and Na_2_O content, but the drying shrinkage of the system increased, and the volume stability worsened. Microscopic analyses showed that with increases in the slag and Na_2_O contents, the hydration products of AAMS composite cementitious materials increased, and C-A-S-H and N-A-S-H filled each other so that the internal structures of AAMS composite cementitious materials were denser, and the porosities were significantly reduced. By comparing and analyzing the Menger sponge model and the fractal model based on the thermodynamic relationships, it was found that the fractal model based on the thermodynamic relationship better reflected the pore size distribution over the whole pore size determination range, and the correlation coefficients R^2^ were above 0.99, indicating that the fractal dimension calculated by the fractal model based on the thermodynamic relationship provided a comprehensive evaluation index for the pore structure characteristics of AAMS composite cementitious materials, and the fractal dimension correlated well with the pore structure parameters, compressive strengths, and drying shrinkage rates of cementitious materials.

## 1. Introduction

Alkali-activated cementitious materials are inorganic cementitious materials with Si-O-Al network structures, which are formed by minerals rich in active aluminosilicate materials (such as metakaolin, slag, fly ash, and steel slag) under catalysis with alkali activators (caustic alkali, alkali-containing silicate, etc.) [1,2]. Compared with ordinary Portland cement, alkali-activated cementitious materials exhibit higher durability and lower environmental load [3]. Therefore, the preparation of concrete with the cementitious materials is considered one of the important directions for sustainable energy savings and emission reductions in the civil engineering industry. The main hydration product of alkali-activated metakaolin cementitious material is N-A-S-H. Alkali-activated metakaolin cementitious materials have good volumetric stabilities and durabilities but have disadvantages such as long setting times and low early strengths [4], and metakaolin is a costly nonrenewable resource with limited reserves [5]. The main hydration products of alkali-activated slag cementitious materials are C-S-H gel and C-A-S-H gel. Alkali-activated slag cementitious materials show high early strength and rapid setting and hardening, and slag is a secondary product and is easy to obtain [6], but it has disadvantages such as large shrinkage of the hardened body [7]. Therefore, the use of slag to replace some of the metakaolin used to prepare alkali-activated composite cementitious materials is a reasonable and economical solution. Moreover, the cementitious materials prepared by using metakaolin slag as a raw material exhibit complementary features of the two raw materials and give full play to the advantages of each raw material by producing a cementitious material with excellent mechanical properties and good volumetric stability to replace ordinary silicate cement.

Recognizing and grasping the fluidity, setting time, compressive strength, and volume stability of alkali-activated cementitious materials is a necessary prerequisite for realizing the application of alkali-activated cementitious materials in civil engineering, while the type and content of the solid powder, the nature and dosage of the alkali activator, and the composition and pore structure of the hydration products are important factors affecting the performance of a cementitious material. For example, Nath et al. [8] and Saha et al. [9] found that the setting time for a fly ash geopolymer was shortened, and the compressive strength was increased when slag powder was incorporated into the fly ash geopolymer. Similarly, Marcin et al. [10] and Song et al. [11] found similar phenomena when they studied fly ash geopolymer mortar. Yip et al. [12] found that the strength of an alkali-activated geopolymer first increased and then decreased with increasing metakaolin content, and the optimum content was 80%. Yang et al. [13] concluded that the fly ash–slag composite system had the highest strength and the densest pore structure with a 50% slag content. Ma et al. [14] concluded that slag effectively improved the physical and mechanical properties of metakaolin-based geopolymers and shortened their setting times when the slag content was 20–30%. Luo et al. [15] controlled the mass fraction of CaO in the system by adjusting the mass ratio of slag to metakaolin and found that increasing the calcium content improved the density of the geopolymer, and that the shrinkage rate was controlled at 4%.

Gao et al. [16] found that the setting times for metakaolin-based geopolymers increased with increasing activator modulus, while the porosity decreased with increasing activator modulus. Lyu et al. [17] suggested that the strengths of geopolymers were inversely related to the activator modulus. In contrast, Alonso et al. [18] and Palomo et al. [19] suggested that the activator concentration was the key factor affecting geopolymer performance. A higher activator concentration resulted in a higher pH for the alkali solution and slower polymerization of the geopolymer in the highly alkaline environment, which degraded the mechanical properties of the geopolymer. Yang et al. [20] concluded that a change in the activator concentration played a decisive role in determining the composition of the products, the extent of the morphological order, and the performance of the product. Fu and Zhan et al. [21,22] found that higher concentrations of activator produced more drying shrinkage and autogenous shrinkage of alkali-activated cementitious materials and reduced the volume stabilities of cementitious materials. Puligilla et al. [23] believed that free calcium prolonged the dissolution of fly ash and enhanced the formation of the geopolymer gel, thus improving the strength of the geopolymer concrete. Kumar et al. [24,25] analyzed the hydration products of fly as–slag geopolymer powders by X-ray diffraction and found that the composite system with fly ash–slag powder did not lead to formation of new cementitious bodies.

Yang et al. [26] compared and analyzed the Menger sponge model, space-filling model, pore axis fractal model, and fractal model based on thermodynamic relationships and found that the fractal model based on thermodynamic relationships described the fractal dimension of the pore structure in geopolymer mortar well, and incorporation of the slag powder into fly ash geopolymers improved the pore structures of the geopolymers. Jiang et al. [27] found that the fractal dimension of the pore structure of alkali-activated slag powder cementitious materials ranged from 3.08 to 3.90 with correlation coefficients between 0.8 and 0.98. The fractal dimensions of the pore structures in the specimens increased with increasing curing age, and the pore structures and spatial distributions became complex. Bai et al. [28] found that the porosity, average pore size, most likely pore diameter, and median pore size of alkali-activated slag mortars decreased with increasing age and alkali content. For the same alkali activator, the fractal dimension of the alkali-activated slag mortar increased with increasing alkali content; for different alkali activators with the same alkali content, the influence of activators on the fractal dimension of alkali-activated slag mortar at 28 d of age decreased in the order water–glass > sodium sulfate > sodium hydroxide.

Based on the above findings, most studies have been conducted on alkali-slag, alkali-fly ash, and their composites. However, there are few quantitative analyses of the macroscopic properties and microscopic pore structure complexities of alkali-activated metakaolin–slag (AAMS) composite cementitious materials. Therefore, in this study, the effects of slag content and Na_2_O content on the macroscopic properties and microstructures of AAMS composite cementitious materials were investigated, and the macroscopic properties and microstructures of AAMS composite cementitious materials were analyzed with a combination of X-ray diffraction (XRD), Fourier transform infrared spectroscopy (FT-IR), scanning electron microscopy (SEM-EDS), and mercury intrusion porosimetry (MIP). The compositions, microscopic morphologies, and pore structure fractal characteristics of the hydration products from composite cementitious materials were analyzed. Finally, the fractal dimension of the cementitious material was solved based on the fractal model of the thermodynamic relationships, and the relationships between fractal dimension, pore structure parameters, compressive strength, and drying shrinkage of the AAMS composite cementitious material were explored.

## 2. Materials and Methods

### 2.1. Materials

#### 2.1.1. Metakaolin and Slag

In this study, metakaolin was used as the main binder. The metakaolin was produced by the Inner Mongolia Chao Brand Company, Hohhot, China, and was a highly active amorphous aluminosilicate material obtained by calcining kaolin-containing clay at a suitable temperature (600–900 °C). Its density and specific surface area were 2300 kg·m^−3^ and 800 m^2^·kg^−1^, respectively. The slag used in this study was finely ground granulated blast furnace slag. Slag was used as the composite binder in this study. The slag was produced by the Ningxia Iron and Steel Group in Zhongwei City, China; it is a byproduct of the blast furnace iron-making process and mainly consists of calcium aluminosilicate material. Its density and specific surface area were 2890 kg·m^−3^ and 410 m^2^·kg^−1^, respectively. The chemical composition and particle size distribution of metakaolin and slag were determined by X-ray fluorescence (XRF) and with a laser particle size analyzer, respectively. The chemical composition and particle size distribution are shown in Table 1 and Figure 1, respectively.

#### 2.1.2. Alkaline Activator

The alkaline activator was a solution consisting of sodium silicate, solid sodium hydroxide, and water. The water–glass solution used in this study was produced by Linyi Lvsen Chemical Co., Ltd., in China, with a SiO_2_ content of 29.94 wt.%, Na_2_O content of 8.86 wt.%, and water content of 61.9 wt.%. Sodium hydroxide flakes with a purity of 99% were produced by the Ningxia Jinyuyuan Chemical Group Co., Ltd., and they were used to adjust the modulus of the water–glass. The water used was deionized water.

#### 2.1.3. Experimental Sand

Standard sand was used to prepare the AAMS. It was produced by the Xiamen Aisiou Standard Sand Co., Ltd. (Xiamen, China), and it conformed to the specifications of GB/T 17671–2021 for cement mortar strength [29]. Its particle diameter, specific gravity, and fineness modulus were 0.08–2 mm, 1.41, and 2.30, respectively.

### 2.2. Sample Preparation

The preparation of AAMS composite cementitious material was carried out with reference to the standard “Cementitious Sand Strength Test Method”, GB/T17671-2021 [29]. In this experiment, the modulus of the fixed activator (Ms = n(SiO_2_)/n(Na_2_O)) was 1.5. First, sodium silicate solution, solid sodium hydroxide, and deionized water were mixed to prepare alkali activators with Na_2_O mass fractions of 8%, 10%, and 12%, respectively, and aged for 24 h. Then, the slag was mixed into metakaolin according at 0, 10%, 30%, and 50% of the total binder mass and poured into the star mixer for premixing so that the metakaolin and slag were mixed evenly, and then the corresponding mass of the activator was weighed into the mixer to establish a water/binder ratio of 0.45 (i.e., the ratio of water to binder in the activator). Finally, the corresponding mass of standard sand was added according to a binder/sand ratio (the ratio of binder to standard sand) of 1:2 and mixed at low speed for 2 min and then at high speed for 3 min. Then, the mixed mortar was poured into a mold with length × width × height dimensions of 40 mm × 40 mm × 40 mm, and then the mold filled with mortar was placed on the vibration table for 3 min to discharge the air bubbles inside the mortar. After the pouring of the specimen was completed, it was placed in a standard curing box (temperature 20 ± 2 °C, relative humidity > 90%) for 24 h and then demolded and cured to the test age. The mixing proportions of AAMS composite cementitious materials are detailed in Table 2.

### 2.3. Test Methods

#### 2.3.1. Fluidity

Fluidity tests were carried out according to the “test method for fluidity and working time of cement asphalt mortar” [30]. The “funnel method” was used with a volume of 640 mL, a diameter of 70 mm at the top and 10 mm at the bottom, and a height of 450 mm. The sample was prepared by referring to the proportions in Table 2 and the method for sample preparation in Section 2.2 (the same as below). The prepared sample was poured into the conical funnel, and the time taken for mortar flowing out from the funnel was taken as the degree of fluidity.

#### 2.3.2. Setting Time

The setting time of the AAMS composite cementitious material was measured with reference to the “test method for water consumption, setting time, and settlement of cement standard consistency” (GB/T 1346-2011) [31]. The prepared mortar was placed in a curing box to measure its setting time. The initial setting time was measured every 5 min, and the final setting time was measured every 15 min. The initial setting time was when the needle sank to 4 ± l mm from the bottom plate. The final setting time was 0.5 mm when the specimen was immersed. The setting time took on the paste (binder only).

#### 2.3.3. Compressive Strength

The AAMS composite cementitious material strength test was carried out with reference to the standard “cementitious sand strength test method” (GB/T17671-2021) [29]. The prepared specimens were placed in a standard curing box and cured to 3, 7, 14, and 28 d test ages. The compressive strengths of the specimens were tested with a constant stress pressure test using a loading rate of 2.4 kN/s, and the test results were taken as the average of three specimens.

#### 2.3.4. Drying Shrinkage

The drying shrinkage test of the AAMS composite cementitious material was carried out according to the standard “dry shrinkage test method of cement mortar” [32]. The homogeneous mortar was injected into a mold with length × width × height dimensions of 25 mm × 25 mm × 28 mm. The mold was removed 24 h after the test block was formed, and the initial length *L_i_* was measured. Then, it was moved into a standard curing box to cure for 48 h and then moved into a drying shrinkage chamber with a temperature of 20 ± 3 °C and a relative humidity of 65%. The lengths of the 3, 7, 14, 28, and 56 d specimens were measured and recorded as *L_x_*, three specimens were formed for each ratio, and the average value of the three specimens was taken as the drying shrinkage value of the AAMS composite cementitious material. The drying shrinkage rate *L_c_* was calculated as shown in Equation (1).
(1)$$L_{c}=(L_{i}-L_{x})/L_{i}\times 100\%$$

#### 2.3.5. Microstructural Tests

Microstructure tests of cementitious material mortar specimens were performed according to the mix proportions in Table 2. After reaching the test age of 28 d, the mortar specimens were broken, placed in anhydrous ethanol for 3 days to terminate hydration, and then placed in a vacuum drying oven for 24 h. Finally, XRD, FT-IR, SEM-EDS, and MIP studies were performed. A Rigaku D/MAX2500 V X-ray diffractometer (made by Rigaku Japan, with a scan rate of 0.02°/s and a scan angle of 5°–70°) was used to analyze the hydration products. A Tensor 27 infrared spectrometer (Bruker, Billerica, MA, USA; 4000–400 cm^−1^ test range; 2 cm^−1^ resolution) was used for phase analyses with sample particle sizes < 74 μm. A Quanta 200 scanning electron microscope (FEI, Hillsboro, OR, USA; 5 nm resolution; 20–10,000 magnification) was used to observe the specimen microstructure period, and the mineral types and chemical composition characteristics of specific hydration products in the micron range were analyzed by energy dispersive spectrometry (EDS). Finally, an Auto Pore IV 9510 automatic Mercury porosimeter (Nocros Corporation, Norcross, GA, USA; 5 nm–0.34 mm measurement range) was used for pore structure analyses.

## 3. Results and Discussion

### 3.1. Fluidity and Setting Time

The effects of different slag contents and Na_2_O contents on the flow time of the AAMS composite cementitious material are shown in Figure 2, which shows that the flow time of the AAMS composite cementitious material decreased from 35.4 s to 12.5 s with an increase in the slag content from 0 to 50%, which constituted a decrease of 64.69%. With increasing slag content, the flow time of AAMS composite cementitious material decreased, which was consistent with the results of Fu et al. [33]. This may be because metakaolin is a clay mineral with an obvious layered silicate structure, so when the content of metakaolin was higher, it adsorbed a large amount of water, thus significantly hindering mortar flow and increasing the mortar flow time; with the increase in slag content, the mortar viscosity was lower, which led to decreases in the AAMS composite cementitious material flow times.

Figure 2b shows that the flow time of the AAMS composite cementitious material decreased from 23.5 s to 16.2 s with an increase in Na_2_O content from 8% to 12%, which constituted a decrease of 31.06%. With the increase in Na_2_O content, the flow time of AAMS composite cementitious material decreased, which was consistent with the results of Lin et al. [34]. The reason is that the higher the Na_2_O content is, the stronger the alkalinity of the solution and the more readily metakaolin and slag will dissolve, which leads to a reduction in the flow time of the AAMS material.

The effects of different slag contents and Na_2_O contents on the setting times of the AAMS materials are shown in Figure 3, which shows that the initial setting time of the AAMS material decreased from 181 min to 63 min, and the final setting time decreased from 273 min to 96 min when the slag content was increased from 0 to 50%, so the initial and final setting times decreased by 65.19% and 64.84%, respectively. The setting time of the AAMS material decreased with increasing slag content, which was consistent with the results of Ma et al. [14]. This was due to the increased slag content; as more Ca^2+^ was introduced, formation of the C-S-H gel and C-A-S-H gel under the action of Ca^2+^ became faster, and the setting time was shorter. Ye et al. [35] found that calcium-containing substances had a procoagulant effect on alkali-activated cementitious materials because Ca^2+^ caused the generated Ca(OH)_2_ to crystallize and precipitate rapidly; this induced the rapid formation of C-A-S-H gels, with the Ca(OH)_2_ crystals serving as the nucleation matrix, which led to shortening of the coagulation time of alkali-activated cementitious materials.

Figure 3b shows that the initial setting time of the AAMS material increased from 57 min to 121 min, and the final setting time increased from 88 min to 175 min when the Na_2_O content was increased from 8% to 12%, and the initial and final setting times increased by 112.28% and 98.86%, respectively. With increasing Na_2_O content, the setting time of the AAMS material increased, which was consistent with the results of Guo [36]. The reason is that within the AAMS composite cementitious material system, the Ca-O bonds and Mg-O bonds were weaker than the Si-O bonds and Al-O bonds, so during the hydration process, the rates of Ca^2+^ and Mg^2+^ dissolution were faster than those of Si^4+^ and Al^3+^; an aluminum-rich film layer was formed quickly on the surface of the solid precursor slag, which prevented further hydration of the composite system and reduced the hydrolysis rates of metakaolin and slag. When the alkali content was increased, the OH^–^ concentration in the mortar increased, the Ca^2+^ dissolution rate increased, and the hydration product film formed on the surface of the solid precursor was denser, so the setting time of the composite cementitious material increased [37,38]. However, Yang [39] found that the setting times of alkali-activated slag cementitious materials decreased with increasing Na_2_O content; Dodiomov [40] and Xu et al. [41] found that the setting times of alkali-activated cementitious materials tended to decrease and then increase with increasing Na_2_O content. With increases in the Na_2_O content, the results of different scholars differed, and this may be related to the reductions in concentrations and raw materials.

### 3.2. Compressive Strength

The effects of different slag contents and Na_2_O contents on the compressive strengths of the AAMS composite cementitious materials are shown in Figure 4. Figure 4a shows that the compressive strengths of AAMS composite cementitious materials with 10%, 30%, and 50% slag content increased by 10.20%, 26.05%, and 32.71%, respectively, after aging for 3 d compared with that of slag at time 0. At the age of 28 d, the compressive strengths of AAMS composite cementitious materials with 10%, 30%, and 50% slag content had increased by 14.60%, 19.59%, and 20.57%, respectively, compared with that of slag at time 0. The compressive strengths of AAMS composite cementitious materials increased with increasing slag content, which was consistent with the results of Zhan [22] and Paulo et al. [42]. The reason is that more slag content produced more C-S-H or C-A-S-H gels with low Ca/Si via hydration of the slag, and these cementitious products were filled in the cracks and pores of the AAMS composite cementitious materials, which made the structure of the composite system denser and thus stronger. However, Cui et al. [43] found that the compressive strengths of alkali-activated metakaolin geopolymers increased and then decreased upon increasing slag content from 0 to 100%. The maximum compressive strength of the geopolymer was found for an age of 28 d and a 40% slag content.

Figure 4b shows that the compressive strengths of AAMS materials with Na_2_O contents of 10% and 12% increased by 39.98% and 59.44% at age 3 d, respectively, compared with that for a Na_2_O content of 8%; at age 28 d, the compressive strengths of AAMS materials with Na_2_O contents of 10% and 12% increased by 18.95% and 24.37%, respectively, compared with that for a Na_2_O content of 8%. The compressive strength of AAMS composite cementitious material increased with increasing Na_2_O content, which was consistent with the results of Timakul et al. [44]. The reason for this is that when the Na_2_O content was higher, the alkali solution contained a large amount of OH^−^, and the solution pH increased, which accelerated breakage of the Si-O-Si, Si-O-Al, and A1-O-Al bonds in the silica–alumina material; this facilitated the reconstruction and reactions of the AAMS composite cementitious material, thus increasing the compressive strength. Similarly, Bian et al. [45] found that the compressive strength of alkali-activated slag/fly ash porous concrete increased with the increase of alkali content.

### 3.3. Drying Shrinkage

The effects of different slag contents and Na_2_O contents on the drying shrinkage of the AAMS materials are shown in Figure 5. Figure 5a shows that drying shrinkage of the AAMS composite cementitious material was smallest when the slag content was 0. At the age of 28 d, the drying shrinkage of the AAMS material with 50% slag content had increased by 215.11% compared with that for a slag content of 0. The drying shrinkage of the AAMS material increased with increasing slag content, and after the age of 28 d, the drying shrinkage showed a stable trend with small increases, which was consistent with the results of a previous study [7,12,46]. This was due to the increased Ca^2+^ introduced by the increase in slag content, and the addition of Ca^2+^ prompted an increase in the hydration product, C-A-S-H gel, of the cementitious material, which resulted in a lower creep modulus due to the viscoelastic–viscoplastic behavior of the C-A-S-H gel. Binding of the alkali ions also reduced the regularity of the stacking structure, making it easier to disintegrate and redistribute under drying conditions, so there was more shrinkage [47,48].

Figure 5b shows that at the age of 28 d, the drying shrinkage of the AAMS composite cementitious materials with Na_2_O contents of 10% and 12% had increased by 79.85% and 129.40%, respectively, compared with a Na_2_O content of 8%. With the increase in Na_2_O content, the drying shrinkage increased, which was consistent with the results of Fu et al. [21]. This occurred because as the Na_2_O content was increased, the OH^–^ concentration in the activator solution increased, and the metakaolin and slag particles dissolved more quickly to release ions to react with the large number of [SiO4 ]^4−^ ions provided by the water–glass solution; this consumed water faster to form a gel network system to generate capillary pressure, which resulted in increased system shrinkage. At the same time, the ions that dissolved quickly in the pore solution reduced the water activity, which led to decreases in the internal relative humidity [49]. In addition, the drying shrinkage of the AAMS composite cementitious materials was also related to the nature of the reaction product [50] and the pore structure [51]. Therefore, in Section 3.4, Section 3.5, Section 3.6, Section 3.7, we will discuss the results of drying shrinkage of the AAMS composite cementitious materials with microstructural analyses.

The relationship between compressive strength and drying shrinkage of the AAMS composite cementitious materials is shown in Figure 6. Figure 6a,b show that the compressive strength drying shrinkage of the AAMS material increased with increasing slag content and Na_2_O content. This result is consistent with that of Zhan et al. [22]. However, from Figure 6a,b, it can be seen that at the age of 3 d, with the increase of slag content and Na_2_O content, the compressive strength and drying shrinkage of AAMS composite cementitious material increased slightly; at 28 d, with the increase of slag content and Na_2_O content, the compressive strength and drying shrinkage of AAMS composite cementitious material increased greatly. This is consistent with the research of Li [52] and Zhan [53]. The reason may be that with the increase of age, the hydration of AAMS composite cementitious material is more complete, and more hydration products are generated, which leads to the increase of compressive strength and drying shrinkage of AAMS composite cementitious material. The slag content and Na_2_O content increased the compressive strength of the AAMS material; however, they also led to increases in drying shrinkage.

### 3.4. XRD

The effects of different slag contents and Na_2_O contents on the XRD data for AAMS composite cementitious materials are shown in Figure 7. The curves for the AAMS composite cementitious material all showed amorphous dispersion peaks, and the positions of the main dispersion peaks were basically the same; all of them were between 25° and 35°, and stronger diffraction peaks for the reaction products in the hardened body of the cementitious material indicated that more reaction products were formed. As shown in Figure 7a, the dispersion peak for the AAMS composite cementitious material was near 26° when the slag content was 0. With increasing slag content, the dispersion peak for the AAMS composite cementitious material was obviously shifted to the right; the peak appeared near 29°, and a narrower peak width meant a higher peak. This indicated that the proportion of C-A-S-H gel in the hydration product continued to increase with increasing slag content. On the other hand, although the compositions and contents of the reaction products cannot be accurately judged by XRD analysis alone, the crystalline material consumption can be used to qualitatively analyze the polymerization reaction of the cementitious material. Figure 7a indicates that as the slag content was gradually increased from 0 to 50%, the contents of Ca_2_Mg(Si_2_O_7_), Ca_2_Al_2_SiO_7_, and C-A-S-H gels increased, and the peak for N-A-S-H did not increase significantly, which indicated that the slag promoted the geopolymerization reaction of metakaolin under the action of the alkali activator so that more metakaolin produced geopolymer gels. The reaction product comprised the geopolymer gel and the C-A-S-H gel generated by slag hydration.

Figure 7b shows that when the slag content was 0–50%, the Ca_2_Mg(Si_2_O_7_) and Ca_2_Al_2_SiO_7_ crystallization peaks weakened with increasing Na_2_O content, which may have been because the increase in Na_2_O content promoted hydration of the metakaolin cementitious material, consumed the above mineral components, and generated more C-A-S-H gel. Moreover, Figure 7b also shows that the C-A-S-H gel peak tip tended to move to higher angles with increasing Na_2_O content, and the trend was more obvious with higher Na_2_O contents, while the characteristic peak for the C-A-S-H gel at approximately 30° also gradually increased in intensity.

### 3.5. FT-IR

The effects of different slag contents and Na_2_O contents on the FT−IR spectra of AAMS composite cementitious materials are shown in Figure 8. Absorption peaks were observed at approximately 3443 cm^−1^, 1645 cm^−1^, 1015 cm^−1^, and 447 cm^−1^. The stretching vibration peak of –OH occurred at wavenumber 3343 cm^−1^ and the H-O-H and bending vibration peaks at wavenumber 1645 cm^−1^. This indicates that chemically bound water or generated hydroxyl groups were present in the product gels of the AAMS materials, and these bound waters had properties similar to those of chemically bound water in N-A-S-H gels and C-A-S-H gels. As shown in Figure 8a, absorption peaks at approximately 1015 cm^−1^ were very obvious in all groups, and with increasing slag content in the system, the absorption peaks tended to move in the direction of lower wavenumbers. The absorption peaks at approximately 1015 cm^−1^ represent nonuniform stretching vibrations of Si-O. Usually, the absorption peak shifts from high to low wavenumbers due to doping of the AlO_4_ structure during polymerization, which is recognized as a characteristic peak for the polymerization of inorganic minerals. This indicates that the alkali-activated polymerization reaction occurred with different slag contents. After adding slag to the metakaolin system, the absorption peak moved from a high to a low frequency, indicating that the degree of polymerization decreased, and the increase in slag content influenced the degree of Si-O-Al chain polymerization. The vibrational peak at approximately 447 cm^−1^ represented a symmetric bending vibration of the Si-O-Si functional group, and the intensity gradually decreased with increasing slag content. This is because incorporation of the slag increased the consumption of SiO_2_ in the metakaolin, and this SiO_2_ may also have reacted with Ca^2+^ in the slag to form C-S-H gels, in addition to the N-A-S-H gels formed by geopolymerization of metakaolin.

Figure 8b shows that the infrared peaks for functional group vibrations in the AAMS composite cementitious materials were the same for different Na_2_O contents, and with increases in Na_2_O content, the T-O-Si (T = Si or Al) peak at 1015 cm^−1^ moved to a lower wavenumber, which indicated that the extent of reaction for the AAMS composite cementitious material was higher under highly alkaline conditions; this proved that the alkaline environment favored the reaction of metakaolin and slag. Therefore, with greater Na_2_O content, more hydration products were generated via reactions of the AAMS composite cementitious material, and the compressive strength was increased.

### 3.6. SEM-EDS

The effects of different slag contents and Na_2_O contents on the SEM-EDS data for AAMS composite cementitious materials are shown in Figure 9. The images M1S0(a) and M5S5(a) in Figure 9 show that when the slag content was 0, the products of AAMS composite cementitious materials showed relatively loose morphologies, mainly in the form of a lamellar structure; this is because the AAMS materials of the pure metakaolin group had low reaction levels and generated products with three-dimensional network gels whose structures were close to the structures of zeolites, which led to microstructural loosening. With increases in the slag content, the internal structure of the AAMS composite cementitious material became compact and dense because the addition of slag led to the generation of large amounts of amorphous C-A-S-H gel inside the composite cementitious material; this filled the gaps of the three-dimensional mesh structure and made the internal structure denser, which was beneficial for increasing the density and compressive strength of the AAMS composite cementitious material. However, a large number of cracks appeared in the internal structure of the AAMS material when the slag content was increased to 50%, which may have been related to the sample preparation and drying shrinkage of the C-A-S-H gel. In the polycondensation process generating capillary tension in cementitious materials, shrinkage may be caused by water evaporation and uneven internal pressures. Yip et al. [12] also concluded that a high slag content adversely affected the structure of the cementitious material.

Images M7S3-8(a) and M7S3-12(a) in Figure 9 indicate that when the Na_2_O content was 8%, there was a large amount of unreacted metakaolin inside the AAMS material, and the structure was loose due to the cracks generated. With an increase in the Na_2_O content to 12%, more C-(N)-A-S-H gel was generated by the system, and some of the gel formed a more compact structure, making the structure in the system denser; at the same time, microcracking in the internal structure of the system also increased. This is because more reaction heat was released at higher Na_2_O contents, which led to increased shrinkage and cracking during the rapid hardening process. At the same time, with the increased Na_2_O content, more C-A-S-H gel was generated, and the C-A-S-H gel itself shrank more, which led to shrinkage of the AAMS material and poor volume stability [21]. As shown in Section 3.3, eight points with different characteristics were selected for EDS analyses to further reveal the mechanism of these changes, and the results are shown in Table 3.

Images M1S0(b) and M5S5(b) in Figure 9 and Table 3 confirm that with increasing slag content, the Ca content in the AAMS material system increased significantly, and the Al and Si contents decreased because the increase in slag content introduced more Ca and promoted the reactions of metakaolin cementitious material. Images M7S3-8(b) and M7S3-12(b) in Figure 9 and Table 3 show that the contents of Ca, Al, and Si decreased with increasing Na_2_O content, which was due to the enhanced alkalinity of the solution and generation of more hydration products. The Al/Si ratio in the AAMS composite cementitious material gradually decreased with increasing Ca/Si, indicating that the increase in Ca/Si hindered the combination with aluminum ions. In addition, Na/Si increased slightly with increasing Ca/Si. Qin et al. [54] concluded that flocculent N-A-S-H gels can coexist with C-A-S-H gels when 0 < Ca/Si < 0.6, which is consistent with the data for the M5S5 group, M7S3-8 group, and M7S3-12 group; these data indicated that N-A-S-H and C-A-S-H gels were generated in the system after slag incorporation. However, when Ca/Si ≥ 0.6, the calcium ions did not react completely, which is consistent with the data for point 2 in M5S5(b). The increase in slag content increased the amount of C-A-S-H gels generated, enhanced the influence of calcium ions on the microstructures and chemical compositions of N-A-S-H gels, increased Ca/Si, provided a denser structure of the system, and increased the compressive strength of the system (see Section 3.2). With increasing alkali content, Ca/Si and Al/Si in the system decreased, and Na/Si increased. This is because the Na_2_O content increased and introduced more Na ions, so the Na/Si in the system increased, and, at the same time, more Ca, Si, Al, and Na reacted to form N-A-S-H and C-A-S-H gels in the highly alkaline environment, so Ca/Si and Al/Si decreased.

### 3.7. MIP

#### 3.7.1. Comparison of Pore Size Distribution of the AAMS Composite Cementitious Materials

The effects of different slag contents and Na_2_O contents on the pore size distribution of the AAMS composite cementitious material is shown in Figure 10, and the pore structural parameters are listed in Table 4. In Figure 10, the pore size distribution curves of the cementitious materials with 0, 10%, and 30% slag contents were roughly similar in shape, and the slag content did not lead to formation of too many capillary pores and too few gel pores in the cementitious materials. The pore volume of the AAMS composite cementitious material decreased when the slag content was increased to 50%, and the most likely pore diameter was in the range 1–10 nm. When the slag content was 50%, the curve shape for the range of pore sizes in the AAMS composite cementitious material differed from the pore size distribution curve when the slag contents were 0, 10%, and 30%. The peak value of the AAMS material at approximately 10 nm decreased with increasing slag content, which meant that the increase in slag content made the AAMS composite cementitious material less likely to exhibit the maximum pore size and a more uniform pore structure distribution.

Based on previous studies [55], the different pores were classified into the following categories: gel pores (<10 nm), transition pores (10–100 nm), capillary pores (100–1000 nm), and macropores (>1000 nm). The distribution of pore volume showed that the pores of the AAMS composite cementitious materials were mainly distributed in the range of 10 nm–100 nm. With an increase in slag content, the number of transition pores of the AAMS material decreased, and the number of gel pores increased. When the slag content was 50%, the proportion of transition pores in the AAMS material decreased by 57.28% compared with a slag content of 0, while the gel pores increased by 446.85%. As seen from Table 4, the total pore volume and porosity of the AAMS material showed decreases with increasing slag content, and this was consistent with the study of Yang et al. [26]. It was further suggested that incorporation of slag generated more C-A-S-H gels, which refined the pore structure of the cementitious material, made the pore distribution more uniform, and increased the compressive strength of the cementitious material. However, the drying shrinkage of the AAMS composite cementitious materials increased due to the substantial shrinkage of the C-A-S-H gels themselves [56].

With increases in the Na_2_O content, the pore size distribution curve for the AAMS composite cementitious material retained similar shapes, and the most likely pore diameter for the AAMS material was approximately 11 nm for Na_2_O contents of 8% and 10%, and the most likely pore diameter was in the range of 1–10 nm for a Na_2_O content of 12%. The peak value of the pore size distribution curve at approximately 10 nm decreased with increasing Na_2_O content, which indicated that the highly alkaline environment favored a more complete reaction of the AAMS composite cementitious material, resulting in a denser internal structure and a more uniform pore structure distribution. As shown in Figure 10 and Table 4, the number of transition pores in the AAMS material decreased, and the number of gel pores increased with increasing Na_2_O content, and the transition pores of the AAMS material decreased by 4.41% and 31.40% and the gel pores were larger by 13.09% and 53.59% when the Na_2_O content was 8% relative to when the Na_2_O contents were 10% and 12%, respectively. Similarly, the higher Na_2_O content decreased the total pore volume and porosity of the AAMS material, and these results are consistent with those of Zhan et al. [22]. This is because the highly alkaline environment promoted the dissolution and decomposition of metakaolin and slag particles, which led to refinement of the synthetic gel pores, an increase in the proportion of gel pores, and an increase in capillary tension; this generated an increase in the drying shrinkage of the AAMS materials.

#### 3.7.2. Comparative Analysis of the Fractal Dimension of AAMS Composite Cementitious Materials

At present, there are multiple models based on MIP testing to study the fractal dimension of the pore structures of cement-based cementitious materials. The Menger sponge model, space-filling model, pore axis fractal model, and fractal model based on thermodynamic relationships are relatively mature. Among these models, the Menger sponge model is the most widely used. The Menger sponge model, space-filling model, and pore axis fractal model all weaken the pore structure into an ideal mathematical geometry when modeling and then obtain a solution for the fractal dimension. These assumptions may cause differences between the model and the actual pore structure, resulting in calculation deviations. However, the fractal model related to the thermodynamic relationship is based on the principle that an increase in the surface energy of the mercury liquid surface generated by mercury pressure is equal to the work done by the external force on the mercury, and the assumption reached for the pore structure while solving for the fractal dimension is closer to the actual situation, so it may be more suitable than other models for calculating the fractal dimension of the concrete pore fractal dimension of the structure. However, most of the current models for fractal dimensions were developed for studying cement-based materials, and few of them were generated for studying alkali-activated cementitious materials [26,57,58,59]. Therefore, in this study, the Menger sponge model and fractal model based on thermodynamic relationships were used to compare and analyze the fractal dimension of AAMS composite cementitious materials, and the relationships between fractal dimension and porosity, total pore area, average pore size, median pore size, compressive strength, and drying shrinkage for AAMS composite cementitious materials were considered.

The principle of the Menger sponge model is as follows [60]: a cube m with the original side length of *R* is divided equally, and a small cube with the side length of *R/m* is newly generated. After removing *n* of them according to a certain rule, the number of remaining cubes is *m*^3^ – *n*. The pore fractal dimension *D_f_* after *k* iterations of this rule satisfies Equation (2):(2)$$\mathrm{lg}(-\frac{dV}{dD})\propto (2-D_{f})\mathrm{lg}D$$
where *V* is the amount of mercury in the pore, in mL·g^−1^, and *D* is the pore diameter, in nm.

The principle of the fractal model based on the thermodynamic relationship is as follows [59]: when measuring the relationship between pore volume and pore diameter for a porous substance by using the mercury pressure method, the work done on the mercury by the external environment is equal to the increase in surface energy of the mercury liquid entering the pore, so the pressure *p* applied to the mercury and the amount of incoming mercury *V* satisfy Equation (3).
(3)$$\int_{0}^{r}pdV=-\int_{0}^{s}\sigma \cos\theta dS$$
where *σ* is the surface tension of the mercury, in N·m^−1^; *θ* is the contact angle between the mercury and the sample, in degrees; *S* is the surface area of the substance to be measured, in m^2^; and *V* is the hole volume of the object to be measured, in m^3^.

Through dimensional analysis, the fractal scale of the pore surface area *S* of a porous material can be correlated with the pore diameter *D* and pore volume *V* to obtain an expression for the fractal model. For the mercury injection stage, Equation (3) can be approximated as a discrete form, as shown in Equation (4):(4)$$\sum_{i=1}^{n}{\bar{p}}_{i}\Delta V_{i}=Cr_{n}^{2-D_{S}}V_{n}^{D_{S}/3}$$
where ${\bar{p}}_{i}$ is the average pressure of the *i*-th mercury injection, in Pa; ∆*V_i_* is the volume of the *i*-th mercury injection, in m^3^; *n* is the number of times the pressurized mercury was injected; *D_n_* is the diameter of the pore corresponding to the *n*-th mercury injection, in m; *V* is the cumulative volume of the *n*-th pressurized mercury injection, in m^3^; and *C* is a constant.

In this study, let
$$W_{n}=\sum_{i=1}^{n}{\bar{p}}_{i}\Delta V_{i},Q_{n}=V_{n}^{1/3}/D_{n}$$

Then,
(5)$$\mathrm{lg}(\frac{W_{n}}{D_{n}{}^{2}})=D_{S}\mathrm{lg}Q_{n}+\mathrm{lg}C$$

Equations (2) and (5) indicate that calculation of the fractal dimension for the pore structure based on the MIP test can be transformed into the problem of studying the slope of a logarithmic function with respect to the pore volume, pore diameter, and incoming mercury pressure.

The pore structure scatter data obtained by the two mathematical models used in calculating the fractal dimension are shown in Figure 11 and Figure 12. Figure 11 and Figure 12 show that when calculated based on the sponge model, the M1S0, M9S1, and M1S1 groups showed close linear relationships (the square of the correlation coefficient R^2^ > 0.9), and the correlation coefficients of M7S3, M7S3-8, and M7S3-12 were all less than 0.9, which showed that the result of the sponge model was relatively discrete. The significance of the correlation coefficient value as used in this paper is referred to in “MedCalc statistical analysis method and application” [61], as shown in Table 5. The correlation coefficient R^2^ of the pore structure calculated by the thermodynamic model was above 0.99, which meant that the fractal model based on the thermodynamic relationship reflected the pore size distribution over the whole pore size measurement range.

Table 6 shows that the conclusions reached with the two models were basically the same; that is, with increasing slag content and Na_2_O content, the fractal dimension of the AAMS composite cementitious material increased. However, the fractal dimension of the pore structure of the AAMS material in the fractal model based on the thermodynamic relationship was between 2.83 and 2.85, while the values calculated with the Menger sponge model were between 3.2 and 3.4. From the basic concepts of topology and fractal theory [62], the fractal dimension of the specimen was greater than 2.0, which indicated that the pore distribution pattern of the specimen was irregular and complex and could not be described by Euclidean geometry. The larger the fractal dimension, the more complex the pores, which was indicated by the high percentage of gel pores (<10 nm) and transition pores (10–100 nm) in the pore size distribution (see Figure 10). Considering that the fractal dimensions of the pore structures of cementitious materials in general are between 2.0 and 3.0, it can be concluded that the fractal model based on the thermodynamic relationship is better; therefore, in the subsequent study, the fractal model based on the thermodynamic relationship was chosen for further discussion.

In summary, incorporation of the slag powder and the increase in Na_2_O content improved the pore structures of the AAMS composite cementitious materials but also made the pore structure complex, so the fractal dimension increased.

### 3.8. Relationship between the Fractal Dimension and Pore Structure Parameters Based on the Thermodynamic Relational Model

#### 3.8.1. Relationship between Fractal Dimension and Porosity

The relationship between the fractal dimension and porosity of AAMS composite cementitious materials is shown in Figure 13. As shown in Figure 13, the linear regression R^2^ for the fractal dimension and porosity was 0.73537, which showed that the correlation between them was good, and they were negatively correlated; therefore, the denser the internal structure of the AAMS material, the larger the fractal dimension, and the more complex the system structure. The fractal dimension characterized the compactness of the cementitious material and the variations in pore sizes well. Therefore, under certain conditions, the relative porosities of different materials can be inferred by comparing their fractal dimension of the pore volume.

#### 3.8.2. Relationship between Fractal Dimension and Total Pore Area

The relationship between the fractal dimension and total pore area of the AAMS composite cementitious material is shown in Figure 14. The larger the total pore area of the AAMS material, the greater the number of pores with small pore sizes or the greater the roughness of the pore surfaces; that is, the pore area reflects the amounts of pore sizes to a certain extent. As seen from Figure 14, the R^2^ value of the linear regression of the fractal dimension and total pore area was 0.62522, which showed that the correlation between fractal dimension and total pore area was weak, and the fractal dimension of the cementitious material decreased with increasing total pore area. This result is contrary to the results of a study on the pore structure of cement mortar reported by Jin et al. [57]. The reason may be that the AAMS composite cementitious material had to be activated with alkali, which led to a pore structure more complex than that of cement mortar, thus leading to a decrease in the total pore area of the AAMS material with increasing fractal dimension.

#### 3.8.3. Relationship between Fractal Dimension and Average and Median Pore Sizes

The average pore size and median pore size (by volume) are parameters used to characterize the average pore size and reflect the distribution of pore sizes when the porosity is held constant. When the porosity is constant, the larger the average pore size and median pore size are, the larger the number of large pores in the pore structure, and vice versa. The relationships between the fractal dimension and the average pore diameter and median pore diameter of the AAMS composite cementitious materials are shown in Figure 15. As shown in Figure 15, the R^2^ value of the plots of fractal dimension versus average pore diameter and the median pore diameter of AAMS composite cementitious materials were 0.65626 and 0.56997, respectively, which indicated that the correlations between fractal dimension and average pore diameter and median pore diameter were weak; with increasing fractal dimension, the average pore diameter and median pore diameter of the cementitious materials were reduced.Has been checked.

### 3.9. Relationship between Fractal Dimension and Compressive Strength and Drying Shrinkage

#### 3.9.1. Relationship between the Fractal Dimension and Compressive Strength

Compressive strength is an important index with which to characterize the mechanical properties of cementitious materials. Figure 16 shows that there was a good positive correlation between the AAMS composite cementitious materials and the fractal dimension. With increases in the slag content and Na_2_O content, the fractal dimension and compressive strengths of the AAMS composite cementitious materials increased, and the correlation coefficients R^2^ were 0.75323 and 0.95572, respectively. The correlation coefficients show that when the Na_2_O content was varied, the correlation between compressive strength and the fractal dimension of the AAMS composite cementitious material was higher, indicating that the pore structure of the cementitious material was more complex when Na_2_O was varied. The complexity of the pore structure was the main factor affecting the compressive strength of the AAMS composite cementitious material. Figure 16 shows that the compressive strength of the AAMS composite cementitious material increased with increases in the fractal dimension, and this result is consistent with the results of a study on the compressive strengths of cement mortars reported by Li et al. [63] This shows that the fractal dimension can characterize the changes in compressive strengths of cementitious materials effectively.

#### 3.9.2. Relationship between the Fractal Dimension and Drying Shrinkage

Drying shrinkage is an important index affecting the volumetric stabilities of AAMS composite cementitious materials. Figure 17 indicates that there was a strong positive correlation between the AAMS composite cementitious material and the fractal dimension. With increasing slag content and Na_2_O content, the fractal dimension and drying shrinkage of the AAMS composite cementitious material increased, and the correlation coefficients R^2^ were 0.93855 and 0.99869, respectively. The correlation coefficient indicated that the fractal dimension had a strong correlation with drying shrinkage, and because the fractal dimension was closely related to the porosity, pore sizes, and pore distribution of the AAMS composite cementitious materials, it also showed that the fractal dimension can quantitatively characterize the complexity of the pore structure. The larger the fractal dimension is, the wider the pore size distribution and the more complex the pore structure. The fractal dimension can make pore structure measurement results more intuitive and comparable, and quantification of pore structures is convenient for further in-depth analyses of the relationship between pore structure and macroscopic performance. Therefore, in seeking to improve the mechanical properties of the AAMS composite cementitious material, the drying shrinkage of the AAMS material can be improved by adjusting the pore structure, which is conducive to further in-depth analyses of the relationships between pore structure and compressive strength and drying shrinkage and provides a theoretical basis for the promotion and application of alkali-activated cementitious materials.

## 4. Conclusions

In this study, macroscopic properties, microscopic analyses, and studies of the fractal dimension for AAMS composite cementitious materials were used to reach the following conclusions:(1)Increasing the contents of slag and Na_2_O improved the workability and mechanical properties of AAMS composite cementitious materials. With increasing slag content, the flow time and setting time of the AAMS composite cementitious material decreased, and the compressive strength and drying shrinkage increased; with increasing Na_2_O content, the flow time of the AAMS composite cementitious material decreased, the setting time increased, and the compressive strength and drying shrinkage increased.(2)When the slag content was 0, the hydration product of the AAMS composite cementitious material was N-A-S-H. With increasing slag content, the proportion of C-A-S-H gel in the composite system increased; at this time, the hydration products in the composite system were mainly N-A-S-H and C-A-S-H. The microscopic morphology showed that C-A-S-H and N-A-S-H filled each other, which made the structure denser and improved the compressive strength of the AAMS composite cementitious material. With increasing Na_2_O content, the degrees of hydration of the solid precursors of the AAMS composite cementitious material were increased, more hydrated substances were generated, the system underwent pore refinement, and the porosity decreased, which led to increased drying shrinkage of the AAMS composite cementitious material.(3)By comparing and analyzing the Menger sponge model with the fractal model based on the thermodynamic relationship, it was found that the fractal model based on the thermodynamic relationship better reflected the pore size distribution over the entire pore size determination range, and the correlation coefficients R^2^ were above 0.99, while dispersion with the Menger sponge model was relatively large. The fractal dimension based on the thermodynamic relationship ranged from 2.83 to 2.85, and the fractal dimension of the Menger sponge model ranged from 3.2 to 3.4. The fractal dimension of both models was greater than 2.0, which indicated that increasing slag and Na_2_O contents made the pore distribution morphologies of AAMS composite cementitious materials irregular and complex.(4)Use of the fractal dimension based on the thermodynamic relationship as a quantitative parameter indicating the pore structure complexity effectively characterized the relative relationships between parameters such as total pore area, average pore size, and median pore size among different pore structures. Therefore, the fractal dimension is a comprehensive parameter with which to evaluate the pore size distribution, which describes the pore size distributions of AAMS composite cementitious materials more accurately than other parameters.(5)The compressive strength, drying shrinkage, and fractal dimension of the AAMS composite cementitious material were strongly correlated, indicating that the complexity of the pore structure is an important factor affecting the macroscopic properties of AAMS composite cementitious materials. The pore structure can be adjusted by changing the contents of slag and Na_2_O to improve the compressive strength of the AAMS material and reduce drying shrinkage. It is helpful to further analyze the relationships between pore structure and the macroscopic properties of the AAMS materials to provide a theoretical basis for application of the AAMS composite cementitious materials.

## Figures and Tables

**Figure 1 polymers-14-05217-f001:**
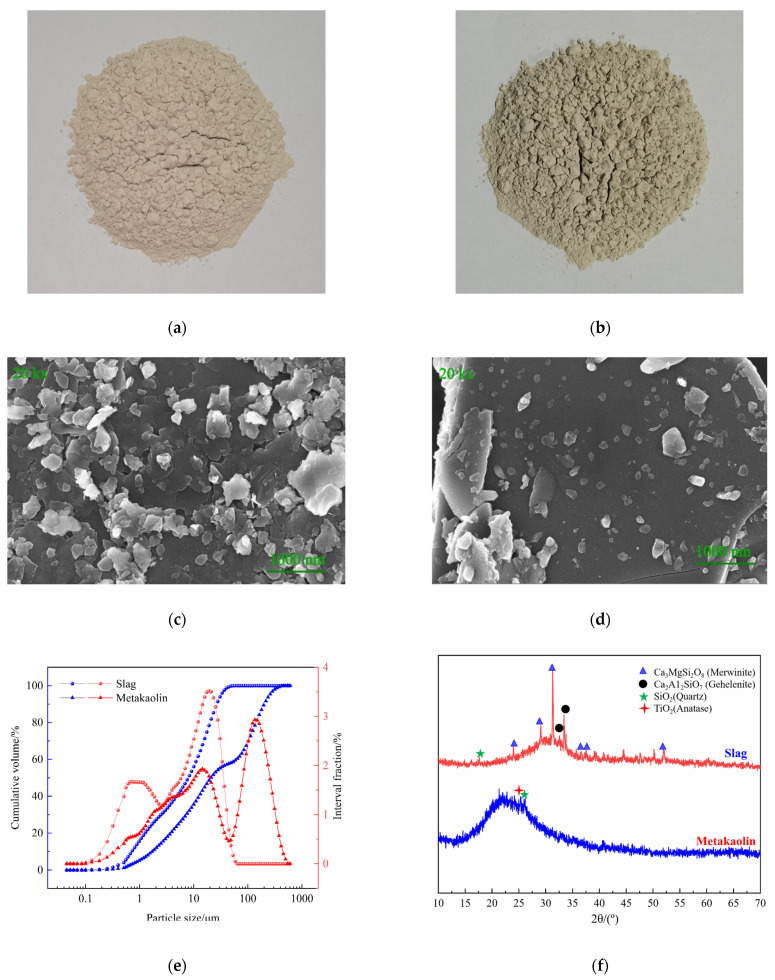
Physicochemical properties of the raw materials. (**a**) Metakaolin. (**b**) Slag. (**c**) SEM image of metakaolin. (**d**) SEM image of slag. (**e**) Particle size distributions of slag and metakaolin. (**f**) XRD results for slag and metakaolin.

**Figure 2 polymers-14-05217-f002:**
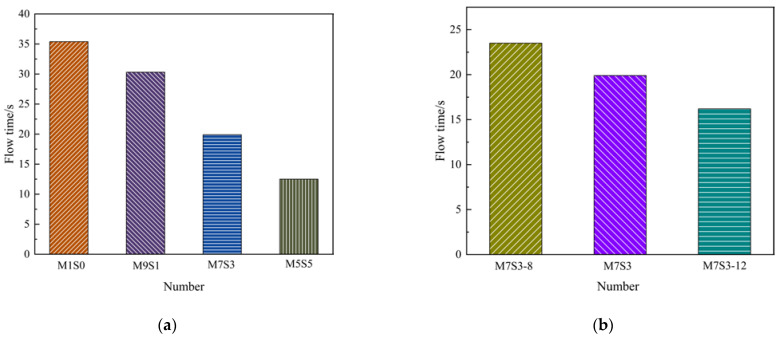
Fluidity of AAMS composite cementitious materials. (**a**) Different slag contents. (**b**) Different Na_2_O contents.

**Figure 3 polymers-14-05217-f003:**
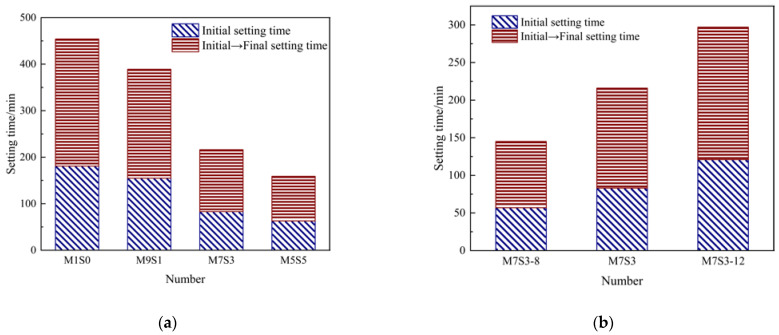
Setting time of AAMS composite cementitious materials. (**a**) Different slag contents. (**b**) Different Na_2_O contents.

**Figure 4 polymers-14-05217-f004:**
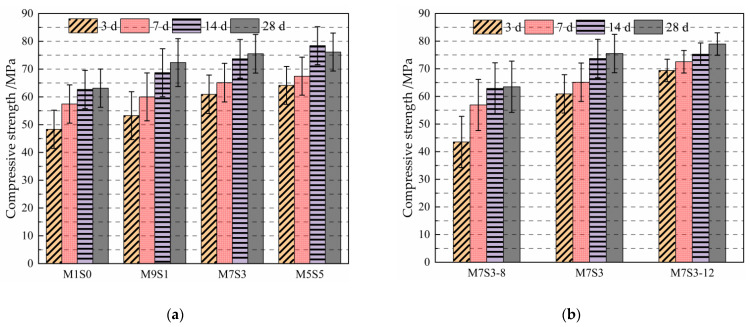
Compressive strengths of AAMS composite cementitious materials. (**a**) Different slag contents. (**b**) Different Na_2_O contents.

**Figure 5 polymers-14-05217-f005:**
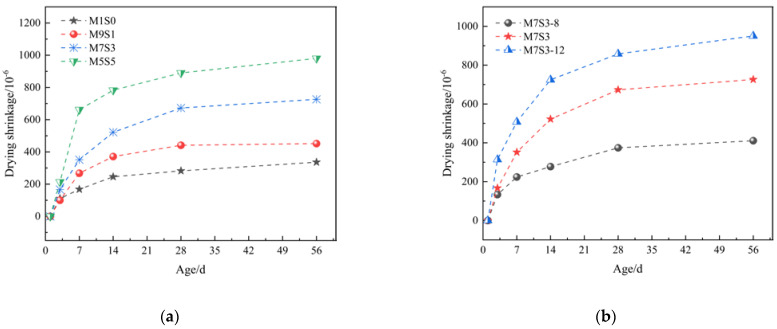
Drying shrinkage of AAMS composite cementitious materials. (**a**) Different slag contents. (**b**) Different Na_2_O contents.

**Figure 6 polymers-14-05217-f006:**
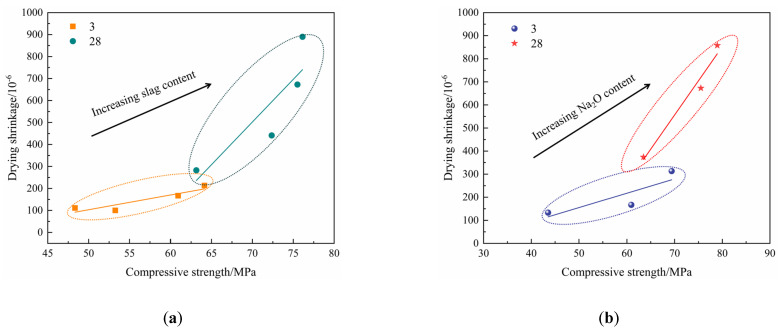
Relationship between compressive strength and drying shrinkage of AAMS composite cementitious materials. (**a**) Different slag contents. (**b**) Different Na_2_O contents.

**Figure 7 polymers-14-05217-f007:**
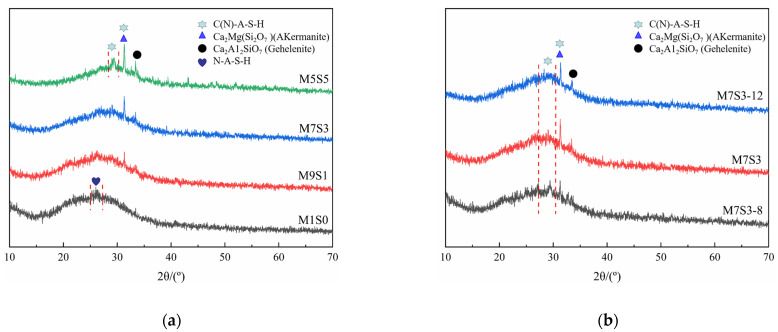
XRD patterns of AAMS composite cementitious materials. (**a**) Different slag contents. (**b**) Different Na_2_O contents.

**Figure 8 polymers-14-05217-f008:**
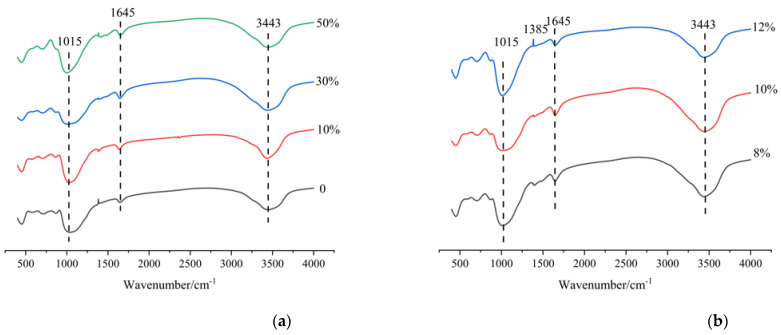
FT−IR spectra of AAMS composite cementitious materials. (**a**) Different slag contents. (**b**) Different Na_2_O contents.

**Figure 9 polymers-14-05217-f009:**
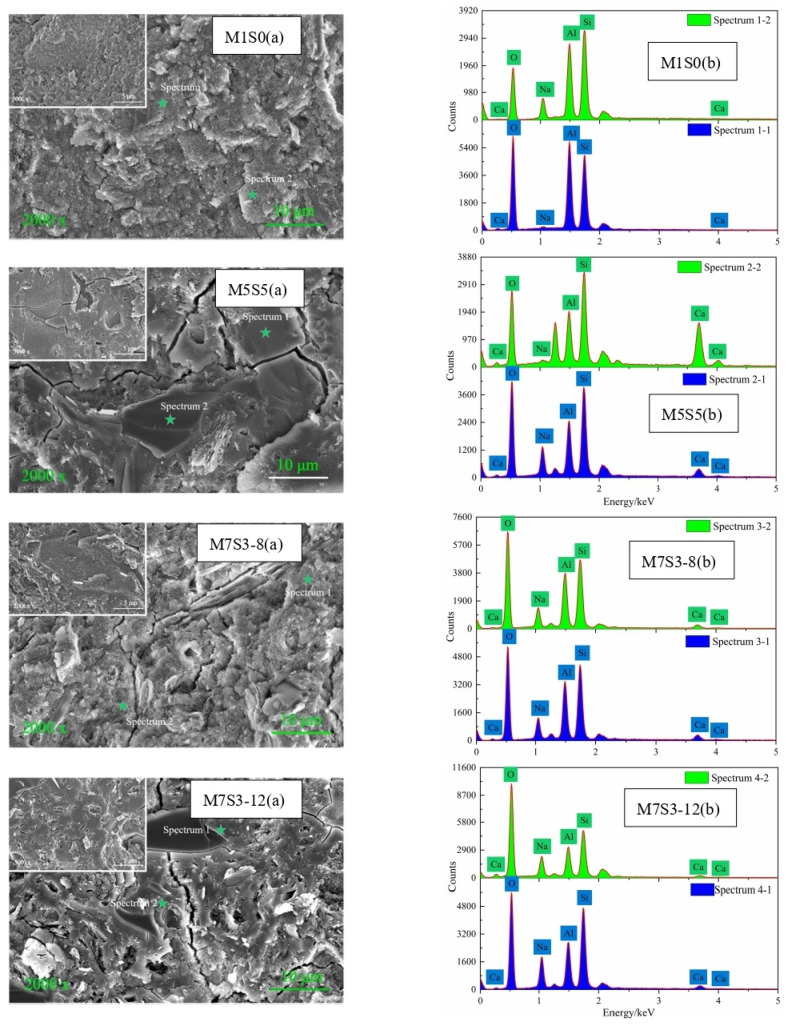
SEM and EDS images of AAMS composite cementitious materials.

**Figure 10 polymers-14-05217-f010:**
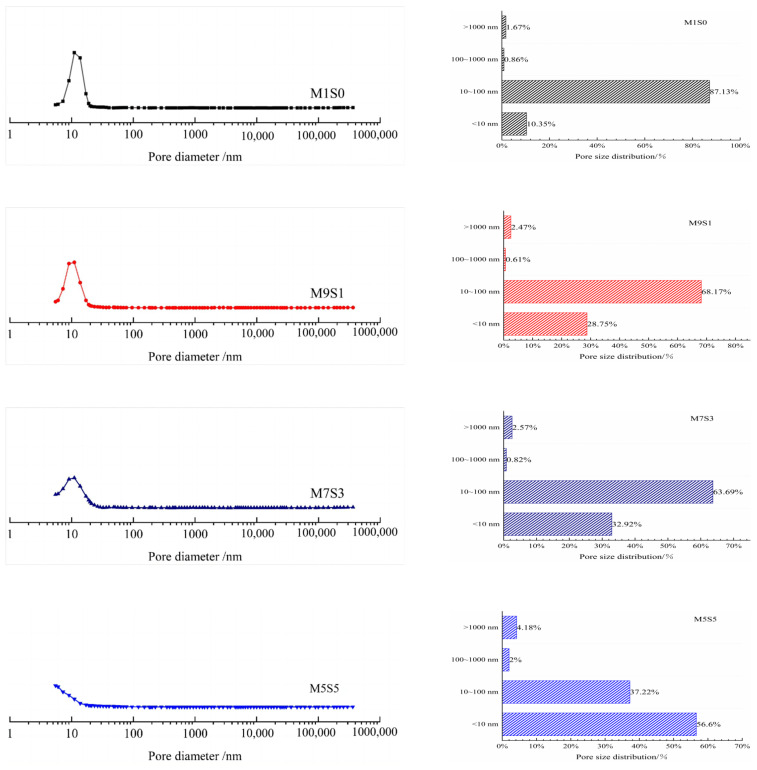
Distributions of pore sizes for AAMS composite cementitious materials.

**Figure 11 polymers-14-05217-f011:**
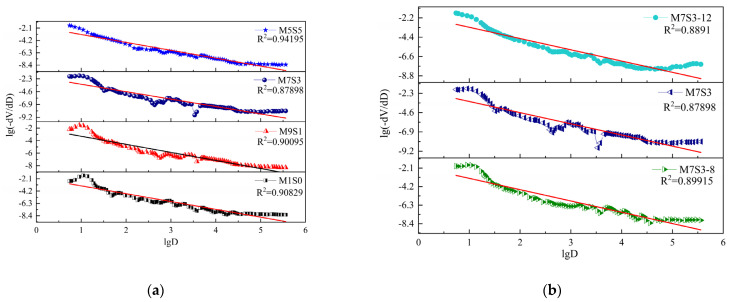
Scatter diagram for the Menger sponge model.(**a**) Different slag contents.(**b**) Different Na_2_O contents.

**Figure 12 polymers-14-05217-f012:**
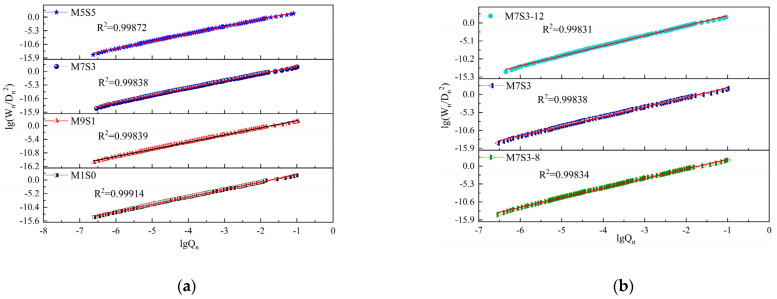
Fractal model scatter plot based on thermodynamic relationships. (**a**) Different Slag contents. (**b**) Different Na_2_O contents.

**Figure 13 polymers-14-05217-f013:**
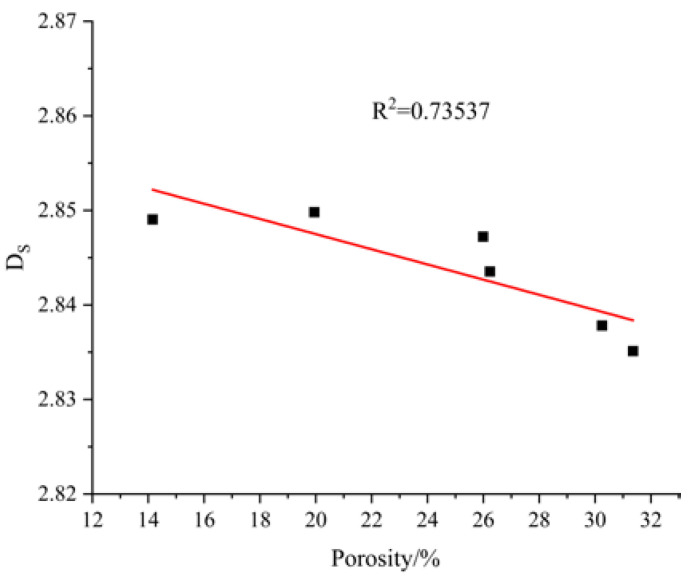
Relationship between fractal dimension and porosity.

**Figure 14 polymers-14-05217-f014:**
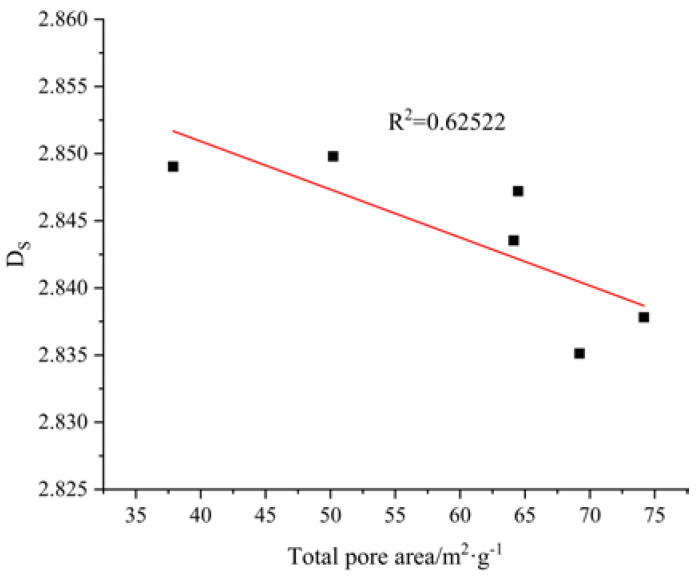
Relationship between fractal dimension and total pore area.

**Figure 15 polymers-14-05217-f015:**
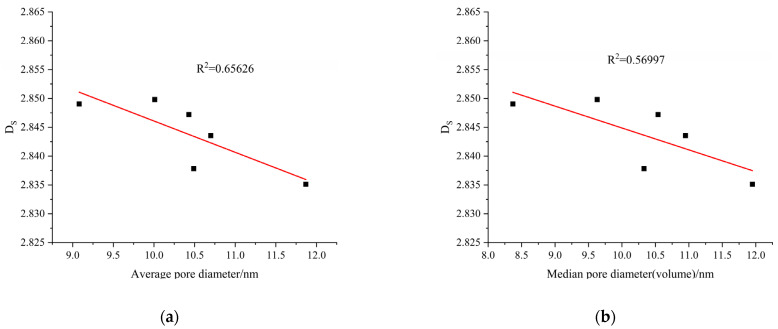
Relationships between the fractal dimension and average pore size and median pore size.

**Figure 16 polymers-14-05217-f016:**
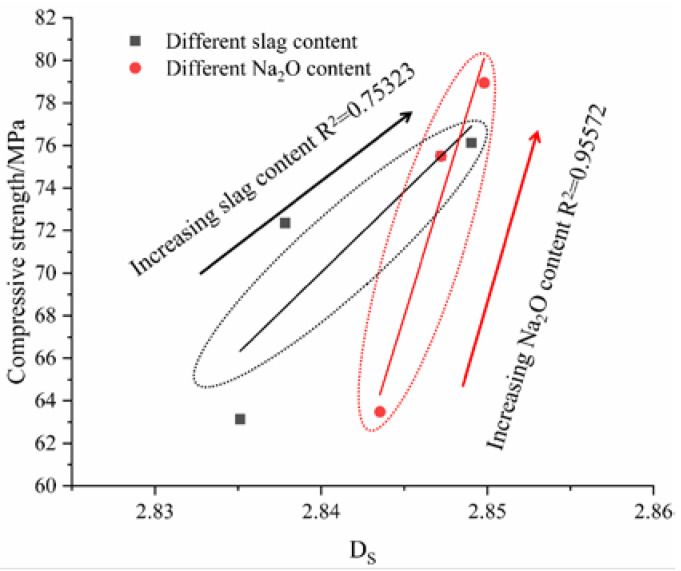
Relationship between the fractal dimension and compressive strength.

**Figure 17 polymers-14-05217-f017:**
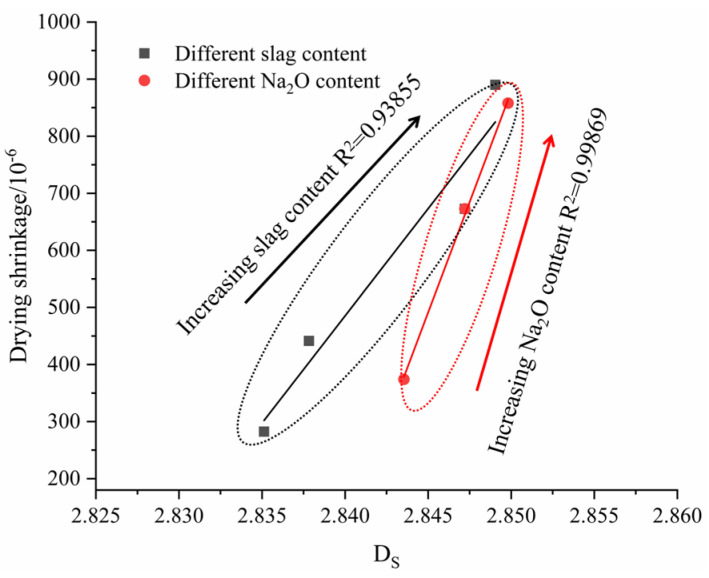
Relationship between the fractal dimension and drying shrinkage.

**Table 1 polymers-14-05217-t001:** Chemical composition and physical index of metakaolin and slag.

Material	Mass Fraction (%)									
	K_2_O	Na_2_O	SO_3_	SiO_2_	Fe_2_O_3_	Al_2_O_3_	MgO	CaO	TiO_2_	LOI
Metakaolin	0.44	0.41	-	49.78	0.93	34.63	2.58	-	1.01	1.1
Slag	0.83	0.73	0.13	35.88	0.46	10.65	11.43	33.54	1.14	1.3

**Table 2 polymers-14-05217-t002:** Proportions of AAMS composite cementitious materials.

Mixtures	Metakaolin (%)	Slag (%)	Na_2_O (%)	Metakaolin (g)	Slag (g)	NaOH (g)	Na_2_SiO_3_ (g)	H_2_O (g)	Sand (g)
M1S0	100	0	10	450	0	32.33	231.15	66.12	900
M9S1	90	10	10	405	45	32.33	231.15	66.12	
M7S3	70	30	10	315	135	32.33	231.15	66.12	
M5S5	50	50	10	225	225	32.33	231.15	66.12	
M7S3-8	70	30	8	315	135	25.86	198.42	93.40	
M7S3-12	70	30	12	315	135	38.79	277.38	38.85	

**Table 3 polymers-14-05217-t003:** Atomic percentages for the elemental composition at each spot in Figure 9 (wt%).

Number	Point	Si	Al	Na	Ca	O	Mg	Al/Si	Ca/Si	Na/Si
M1S0	1	22.63	21.09	0.37	0.01	55.88	0.01	0.93	0	0.016
	2	31.95	21.35	6.25	0.11	40.25	0.10	0.67	0	0.20
M5S5	1	17.29	15.75	6.71	4.32	54.56	1.37	0.91	0.38	0.39
	2	17.19	8.34	0.43	18.78	48.41	6.84	0.49	1.09	0.025
M7S3-8	1	20.53	13.46	5.53	3.16	56.22	1.11	0.66	0.15	0.27
	2	20.01	13.4	5.14	2.24	58.12	0.81	0.67	0.11	0.26
M7S3-12	1	22.36	10.98	8.29	1.88	55.74	0.75	0.49	0.084	0.37
	2	16.94	9.17	7.2	1.48	64.47	0.74	0.54	0.087	0.43

**Table 4 polymers-14-05217-t004:** Pore structure parameters for AAMS composite cementitious materials.

Number	Total Porosity (mL·g^−1^)	Total Pore Area/m^2^·g^−1^	Medium Pore Diameter (V) (nm)	Medium Pore Diameter (A) (nm)	Average Pore Size (nm)	Porosity (%)
M1S0	0.2053	69.202	11.95	11.27	11.87	31.3592
M9S1	0.1945	74.168	10.33	9.63	10.49	30.2487
M7S3	0.1680	64.465	10.54	9.26	10.43	25.9911
M5S5	0.0860	37.872	8.37	7.08	9.08	14.1593
M7S3-8	0.1715	64.127	10.95	9.56	10.70	26.2337
M7S3-12	0.1256	50.205	9.63	7.79	10.01	19.9471

**Table 5 polymers-14-05217-t005:** Significance of the correlation coefficients.

Absolute Value of the Correlation Coefficient	Correlation Strength	Correlation
0.9–1.0	Highly correlated	Correlated
0.7–0.9	Strongly correlated	
0.5–0.7	Weakly correlated	
<0.5	Very weakly correlated	Uncorrelated

**Table 6 polymers-14-05217-t006:** Fractal dimension of AAMS composite cementitious materials.

Number	Menger Sponge Model		Thermodynamic Model	
	R^2^	Fractal Dimension *D_f_*	R^2^	Fractal Dimension *D_s_*
M1S0	0.90229	3.28525	0.9991	2.83512
M9S1	0.89895	3.32061	0.99839	2.83782
M7S3	0.86898	3.33329	0.99838	2.84721
M5S5	0.94111	3.34219	0.9987	2.84904
M7S3-8	0.89115	3.28011	0.99834	2.84354
M7S3-12	0.8811	3.34225	0.99831	2.84981

## Data Availability

The data supporting the findings of this study are available from the corresponding author upon reasonable request.

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
