# Peer review of "Macroscopic Properties and Pore Structure Fractal Characteristics of Alkali-Activated Metakaolin–Slag Composite Cementitious Materials"

_polymers, 2022, doi:10.3390/polym14235217_

Round 1

Reviewer 1 Report

1. The authors observed the properties of materials that are referred as alkali-activated materials, it is not appropriate to use the term cement for these materials, if cement is not used in the mixture.

2. Today time it is not appropriate to call blast furnace slag as a „inexpensive industrial waste“. The term „secondary product“ is used.

3. In the introduction, the authors should describe the formation of hydration products in alkali-activated materials based on metakaolin and granulated blast furnace slag.

4. In the materials chapter, the authors should add that this is finely ground granulated blast furnace slag (ggbfs).

5. The authors should justify why they used right these ratios (metakaolin x slag and content of alkaline).

6. Although the authors have indicated the ratio between binder and sand in the text, it would be appropriate to add the amount of sand used to Table 2.

7. Authors should not use the term paste but mortar when sand is used in the mixture. If necessary, specify which tests were performed on pastes and which on mortars. For example, the determination of the setting time takes place on pastes.

8. Add standard deviations to the results in Figure 4.

9. In the compressive strength and shrinkage section, the authors should elaborate more on the discussion compared to other researches.

10. The authors should mention that some cracks may have occurred only during the preparation of the samples for SEM-EDS analysis (vacuumed) and not by shrinkage.

11. In line 399, the authors should add to the legend what T stands for (T-O-Si).

Author Response

Dear reviewer 1, 

Thank you for your review, I have modified all the questions, please see article and reply, thank you.

Question 1: The authors observed the properties of materials that are referred as alkali-activated materials, it is not appropriate to use the term cement for these materials, if cement is not used in the mixture.

Question 1 has been resolved, please see line 71. In this article, cementitious is not equal to cement.

Question 2: Today time it is not appropriate to call blast furnace slag as a „inexpensive industrial waste“. The term „secondary product“ is used.

Question 2 has been resolved, please see lines 52-53.

Question 3: In the introduction, the authors should describe the formation of hydration products in alkali-activated materials based on metakaolin and granulated blast furnace slag.

Question 3 has been resolved, please see lines 52-53 and 52-54.

Question 4: In the materials chapter, the authors should add that this is finely ground granulated blast furnace slag (ggbfs).

Question 3 has been resolved, please see lines 140-141.

Question 5: The authors should justify why they used right these ratios (metakaolin x slag and content of alkaline).

Question 5: Binder is metakaolin + slag, and metakaolin content + slag content is 100%. Slag, metakaolin and alkali content mainly refer to the following literature calculation. References, e.g.

  1. Zhenming Li, et al. Mitigating the autogenous shrinkage of alkali-activated slag by metakaolin. DOI: 10.1016/j.cemconres.2019.04.016.2.
  2. Bo Fu, et al. Understanding the Role of Metakaolin towards Mitigating the Shrinkage Behavior of Alkali-Activated Slag. Materials 2021, 14, 6962.

Question 6: Although the authors have indicated the ratio between binder and sand in the text, it would be appropriate to add the amount of sand used to Table 2.

Question 6 has been resolved, please see Table 2.

Question 7: Authors should not use the term paste but mortar when sand is used in the mixture. If necessary, specify which tests were performed on pastes and which on mortars. For example, the determination of the setting time takes place on pastes.

Question 7 has been resolved, please see this article.

Question 8: Add standard deviations to the results in Figure 4.

Question 8 has been resolved, please see Figure 4.

Question 9: In the compressive strength and shrinkage section, the authors should elaborate more on the discussion compared to other researches.

Question 9 has been resolved, please see compressive strength and shrinkage section.

Question 10: The authors should mention that some cracks may have occurred only during the preparation of the samples for SEM-EDS analysis and not by shrinkage.

Question 10 has been resolved, please see lines 432-433.

Question 11: In line 399, the authors should add to the legend what T stands for (T-O-Si).

Question 11 has been resolved, please see lines 432-433.

Reviewer 2 Report

Title: Macroscopic properties and pore structure fractal characteristics of alkali-activated metakaolin-slag composite cementitious materials

Overview: The macroscopic and microscopic analyses, as well as the studies of the fractal dimension, were done on the alkali-activated composite cementitious materials incorporated with metakaolin and different contents of slag and Na2O. The paper is well organized and written and the results could be interesting for the readers of the journal. Just some minor comments are as follows:

Comments:

·         In the abstract, please add some quantitative results also.

·         Table 1-Title: of silica fume! (SF was not used in this study)

·         Section 2.3.2 (Line 189): Please check the line that min is correct!

·         Figure 4: Compressive strength: The results were an average of 3 specimens. Please show the standard deviation on the charts.

·         Line 553: it seems a parameter was missed in the beginning.

·         Please check the Title of Fig.15

·         3.9.2: It seems this sub-title should be drying shrinkage (not compressive strength).

Author Response

Dear reviewer 2, 

Thank you for your review, I have modified all the comments, please see article and reply, thank you.

Comment: 1. In the abstract, please add some quantitative results also.

Comment 1 has been resolved, please see lines 19-23.

Comment: 2. Table 1-Title: of silica fume! (SF was not used in this study)

Comment 2 has been resolved, please see line 149.

Comment: 3. Section 2.3.2 (Line 189): Please check the line that min is correct!

Comment 3 has been resolved, please see lines 197-199.

Comment: 4. Figure 4: Compressive strength: The results were an average of 3 specimens. Please show the standard deviation on the charts.

Comment 4 has been resolved, please see Figure 4.

Comment: 5. Line 553: it seems a parameter was missed in the beginning.

Comment 5 has been checked, the article has no missing parameters.

Comment: 6. Please check the Title of Fig.15

Comment 6 has been resolved, please see Figure 15.

Comment: 7. 3.9.2: It seems this sub-title should be drying shrinkage (not compressive strength).

Comment 7 has been resolved, please see 3.9.2 sub-title.

Round 2

Reviewer 1 Report

1. The authors should clearly state why they use the term cementitious for alkali-activated materials. If it is taken from an article, put the citation in quotation marks. The explanation that "cementitious is not equal to cement" is insufficient.

2. Furthermore, the authors should state whether the setting time takes on the paste (binder only) or on the mortar (binder and sand).

Author Response

Dear reviewer, 

Thank you for your review, I have modified all the questions, please see article and reply, thank you.

Question 1: The authors should clearly state why they use the term cementitious for alkali-activated materials. If it is taken from an article, put the citation in quotation marks. The explanation that "cementitious is not equal to cement" is insufficient.

Question 1 has been resolved, please see lines 41-44. The cementitious material generally refers to the material that the powder is mixed with water and has certain gelation. After a certain period of time, condensation or solidification will occur. For example, cement is the most widely used cementitious material. Alkali-activated cementitious material is a new type of inorganic non-metallic cementitious material produced by the reaction of alkaline activator with pozzolanic active or potential hydraulic raw materials. Compared with ordinary Portland cement, alkali-activated cementitious materials generally have the advantages of high strength, rapid strength development, good frost resistance, good acid corrosion resistance and good stability.

Question 2: Furthermore, the authors should state whether the setting time takes on the paste (binder only) or on the mortar (binder and sand).

Question 2 has been resolved, please see lines 199-200. The setting time takes on the paste (binder only).

Round 3

Reviewer 1 Report

Publish as is.